# How Metaphorical Instructions Influence Children’s Motor Learning and Memory in Online Settings

**DOI:** 10.3390/bs15081132

**Published:** 2025-08-20

**Authors:** Weiqi Zheng, Xinyun Liu

**Affiliations:** 1School of Psychology, Beijing Sport University, Beijing 100084, China; 2Laboratory of Sports Stress and Adaptation of General Administration of Sport, Beijing 100084, China

**Keywords:** metaphorical instruction, motor learning, motor memory, children

## Abstract

Metaphorical instructions are widely used in motor skill learning, yet their impact on learning and memory processes in children remains underexplored. This study examined whether metaphor-based language could enhance children’s acquisition and recall of body posture-related motor skills in an online learning environment. Forty-eight children aged 7 to 9 were randomly assigned to receive either metaphorical or explicit verbal instructions while learning 15 gymnastic postures demonstrated through static images. Following the learning phase, participants completed a free recall task, in which they reproduced the learned postures without cues, and a recognition task involving the identification of previously learned postures. Results indicated that children in the metaphor group recalled significantly more postures than those in the explicit group, with no reduction in movement quality. However, no group differences were observed in recognition accuracy or discrimination sensitivity. These findings suggest that metaphorical instructions may enhance children’s ability to retrieve self-generated motor representations but offer limited advantage when external cues are available. The study provides evidence for the value of metaphor-based strategies in supporting immediate motor memory in digital, child-focused learning settings and highlights the potential task-dependency of instructional language effects on memory outcomes.

## 1. Introduction

Motor skill learning refers to the process of acquiring relatively permanent changes in the ability to perform movements, typically through repeated practice and experience ([12]; [20]). Central to this process is motor memory, which involves encoding, storing, and retrieving movement patterns based on bodily states or imagined actions ([22]). Mastery of complex motor skills—such as those found in gymnastics, martial arts, or dance—requires not only the learning of individual actions but also the retention and fluid execution of sequential movements. In this context, motor memory serves as a fundamental cognitive and sensorimotor mechanism that supports athletic performance across various sports, including ball games, swimming, skiing, and so on.

Motor memory transitions from a fragile to a stable state and involves key regions such as the motor cortex and cerebellum ([9]). Studying motor memory in children is particularly important, as it contributes to their motor development, perceptual–motor integration, and broader cognitive growth. Motor memory enables children to store and adapt movement patterns, reflecting the progressive maturation of their motor and neural systems ([3]; [22]). It also facilitates the integration of sensory feedback and motor output, allowing for effective interaction with the environment. The consolidation of motor memory is closely related to brain plasticity. On a practical level, research on motor memory aids in the early identification of developmental disorders and learning disabilities and informs more targeted educational and rehabilitation strategies ([6]). More importantly, mechanisms that enhance motor memory—such as through neural stimulation—offer new intervention approaches for improving learning outcomes in children with special needs ([19]). Therefore, systematic research on motor memory in children not only deepens our understanding of its neurocognitive mechanisms but also provides a solid scientific foundation for developing effective education of motor learning and therapeutic interventions.

Children commonly acquire motor skills through either explicit or implicit learning processes. Explicit learning involves conscious attention to performance and verbalizable rules, typically supported by detailed, step-by-step instructions specifying exact postures or movements ([14]). In contrast, implicit learning occurs with minimal conscious awareness of the underlying mechanics, often resulting in skills that are more robust under stress and dual-task conditions, though learners typically struggle to articulate the rules guiding their performance ([14]). Within this framework, analogical learning represents a distinct form of implicit learning wherein complex movement patterns are conveyed through metaphorical verbal cues, reducing cognitive load by integrating multiple biomechanical parameters into a single, meaningful image or concept ([10]). This learning method is grounded in the theory of embodied cognition, which posits that understanding and memory are rooted in sensorimotor experiences. Analogies help learners bypass the need for extensive declarative processing by engaging intuitive representations of action, making them especially effective for children, whose working memory and executive functions are still developing ([26]). Empirical studies have demonstrated that skills acquired through analogical learning are more resistant to breakdown under pressure, cognitive load, or fatigue compared to explicitly learned skills ([25]).

In the context of pediatric motor learning, analogical approaches have shown promising outcomes across a variety of domains ([1]). For example, in swimming, analogies such as “slice through the water like a knife” help children internalize streamlined posture without the need for technical explanations. In tennis, cues like “brush the ball like painting a wall” intuitively convey the mechanics of topspin generation. In gymnastics, metaphors such as “make your body like a pencil” help reinforce body alignment during landings. These analogies not only simplify biomechanical concepts but also enhance motivation and engagement, leveraging children’s natural affinity for stories, imagery, and play.

Verbal instructions are the most direct means by which learning strategies—whether explicit or analogical—are conveyed during motor skill acquisition. [17] ([17]) showed that both analogy instruction and explicit instruction helped intermediate junior tennis players order mental representations of the tennis serve functionally, potentially leading to better cognitive representations in their long-term memory and better performance over time in terms of accuracy. In addition, a growing body of research has demonstrated that both explicit and analogical instructions can benefit novice learners, with analogical instructions often providing greater advantages in early skill acquisition ([2]; [15]; [16]). For example, [24] ([24]) showed that novice participants performing a hockey push task under analogical instruction exhibited enhanced cognitive efficiency, as indicated by increased EEG alpha power in the left temporal lobe. Similarly, studies in children have shown that analogical instructions can facilitate rapid comprehension and execution of movement sequences ([23]), suggesting that metaphorical language may be particularly well-suited to children’s developmental learning needs. Moreover, recent developmental research emphasizes the importance of aligning instructional strategies with learners’ cognitive and motor readiness. [7] ([7]) analyzed expert opinions on optimal timing for technical skill acquisition and highlighted the need to consider developmental stages in instructional design—an aspect particularly relevant to our child-focused study.

However, most previous research has focused on motor performance and has not addressed how analogical instructions affect different types of memory, such as procedural memory (action reproduction) and declarative memory (movement recognition). Moreover, little is known about how children’s developing linguistic capabilities interact with instructional language to influence motor memory encoding and consolidation. This gap is critical, given that children’s motor learning is heavily influenced by multimodal and language-rich inputs. It should be noted that this study focuses on the linguistic characteristics of verbal instructions used in analogical learning. For clarity and consistency, the term “metaphorical instruction” will be used throughout the remainder of the paper to refer to this form of guidance.

Therefore, the present study addresses this gap by comparing the effects of metaphorical and explicit instructional language on two aspects of children’s motor memory: the ability to reproduce learned body postures and the ability to recognize them in visual tasks. We situate this investigation within the context of online learning, a format that has grown significantly since the COVID-19 pandemic. The use of digital platforms for motor learning introduces certain elements of ecological validity, such as allowing children to learn in familiar home settings. However, the task design remains more controlled and simplified than real-world motor learning. Therefore, the ecological validity of the present study is limited and should be interpreted with caution.

To enhance the ecological validity of analogical learning and better reflect instructional practices in real-world settings, the metaphorical instructions in the present study were not purely metaphorical in form. Instead, we adopted a blended design, where key complex or abstract movement elements were expressed through metaphors, while more straightforward aspects were conveyed explicitly. This hybrid approach aligns with some previous research suggesting that combining implicit and explicit information can enhance motor skill learning ([21]; [27]). Furthermore, using comparable sentence lengths across instruction types helped minimize differences in cognitive load due to working memory constraints, ensuring that performance differences were more likely due to instructional content rather than verbal complexity.

In summary, this study aims to examine whether metaphorical instructions provide measurable advantages in children’s motor memory performance under ecologically valid online learning conditions. This study contributes to the field in three key ways. First, it extends research on analogical instruction to a fully online learning environment, a format increasingly adopted in post-pandemic education yet rarely examined in motor skill acquisition among children. Second, it shifts the focus from moment-to-moment performance accuracy to immediate motor memory outcomes, using both free recall and recognition tasks to assess how different types of instructional language influence memory encoding and retrieval. Third, it investigates these effects in younger children aged 7–9 years, a population often underrepresented in research on motor learning, particularly in digital contexts.

Based on previous findings and theoretical frameworks, we hypothesized that: (1) Children receiving metaphorical instruction will recall a greater number of learned body posture actions than those receiving explicit instruction. (2) No significant difference is expected in the execution quality of recalled actions between the two groups. (3) No significant difference is expected between groups in recognition accuracy or confidence, since recognition tasks rely more on perceptual matching than on embodied or conceptual memory representations.

## 2. Materials and Methods

### 2.1. Participants

Sample size was determined via a priori power analysis using G*Power 3.1 software to achieve 80% statistical power (β = 0.2) at a significance level of α = 0.05. A two-group between-subjects design required at least 22 participants per group. To account for potential attrition in online settings (estimated at 10%), the final sample was expanded to 24 children per group (total *N* = 48). This calculation ensured adequate power to detect medium-to-large effect sizes while maintaining balanced gender distribution.

Forty-eight children (24 girls and 24 boys) aged 7–9 years old, with the consent of their parents, voluntarily attended the study from a lot of cities in China (such as Beijing, Shenzhen, Qingdao, Changchun, and Guilin) through online recruitment. All participants were reported by parents as typically developing, with no known neurological, developmental, or motor impairments. Participants were assigned to the metaphorical or explicit instruction group using a stratified randomization method based on age. Specifically, after recruitment, children were first grouped into age strata to ensure balanced age distribution across conditions. Within each stratum, children were randomly divided into two groups: (a) the metaphorical instruction group (*n* = 24; 13 girls and 11 boys; mean age = 8.2 years old) and (b) the explicit instruction group (*n* = 24; 11 girls and 13 boys; mean age = 8.2 years old). All children received a fixed monetary reward after completing the tasks, regardless of their performance. This approach aimed to sustain engagement without introducing competitive pressure. Nonetheless, the possibility that compensation influenced motivation cannot be entirely ruled out and is acknowledged as a limitation.

The present study was conducted according to the Declaration of Helsinki. The protocol was approved by approved by Sports Science Experiment Ethics Committee of Beijing Sport University.

### 2.2. Materials

To better assess children’s ability to learn and recall motor movements, the present study used 15 images of movement postures modeled after children’s gymnastics as experimental stimuli (see Figure 1). These postures primarily involved the limbs and trunk, were distinct from everyday habitual movements, and were of moderate difficulty for children to perform. The movements were demonstrated by a collegiate gymnast with professional training in gymnastics. To evaluate the familiarity of these postures, 40 undergraduate students from Beijing Sport University were invited to rate each movement on a 5-point Likert scale (1 = very unfamiliar, 5 = very familiar). The average familiarity rating was 1.72 ± 0.87 (M ± SD), indicating that the movements were generally unfamiliar in daily life and thus suitable for the objectives of this study.

The metaphorical and explicit instructions corresponding to each posture were jointly developed by the collegiate gymnast who performed the movements and the authors of this paper (see examples in Table 1). The metaphorical instructions were designed using a blended format. Complex or unfamiliar elements of the posture were described metaphorically, while simpler or structural elements (e.g., “keep feet flat”) were delivered in explicit language. This approach allowed for a clearer understanding and helped ensure that sentence length and verbal load were comparable between conditions, minimizing differences in working memory demands. In addition, to support the recognition task in the study, each target image was paired with a visually similar but distinct posture to serve as a distractor, resulting in a total of 15 distractor images.

Before the formal study, we informally checked the comprehensibility of the metaphorical instructions with 3 children (2 girls, 1 boy) within the target age range (7–9 years). These children were not part of the main sample but matched the study’s demographic criteria. For each instruction, we asked the children to explain their understanding in their own words and to demonstrate the corresponding posture. All 3 children were able to consistently link the metaphorical cues to the intended postures and verbalize the core movement requirements. No major misunderstandings were identified, and no instructions required revision based on this feedback.

### 2.3. Experimental Design and Procedure

This study employed a single-factor between-subjects design, with instruction type (metaphorical vs. explicit) as the independent variable. The dependent variables included the quantity and accuracy of correctly recalled movements during the free recall task, as well as the correct recognition rate, discrimination sensitivity, and confidence level during the subsequent recognition test.

Participants were recruited online and included children aged 7–9 years. After obtaining informed consent from parents, each child participated in a scheduled session via Tencent Meeting, accompanied by a parent. To control variability in the online environment, detailed setup instructions were provided to parents, including how to position the camera to ensure full-body visibility and proper lighting. All sessions were monitored live via video conferencing. The experimenter ensured that each participant was visible and corrected any camera angle or framing issues before beginning. Parental supervision was required throughout the session. All sessions were recorded with consent, and any video with serious technical issues (e.g., excessive lag and obstruction) was excluded from analysis.

During the learning phase, the experimenter presented movement posture images while providing corresponding verbal instructions. Children in the metaphorical instruction group received figurative, image-based explanations, whereas those in the explicit group received literal, step-by-step descriptions. The learning phase consisted of two parts: in the first, children focused on listening and understanding the instructions; in the second, they imitated the actions concurrently while viewing the images.

Following a 10 min rest, children completed a free recall task in which they reproduced as many learned movements as possible in any order, without prompts. With parental permission, all recall sessions were screen-recorded to enable later scoring of movement accuracy. After another 10 min break, children completed a recognition test where they judged whether each presented image was “seen before”, “not seen”, or “not sure”. Monetary compensation was provided upon completion of the experiment. The entire procedure was video-recorded with parental and child consent for subsequent analysis.

The experimental procedure ensured standardized delivery of instructions while enhancing ecological validity through remote, real-world implementation. The task sequence—free recall (targeting procedural memory) followed by recognition (targeting declarative memory)—helped maintain the distinct cognitive demands of each task, while the 10 min intervals reduced cross-task interference and mimicked natural learning rhythms. Screen-recorded data enabled precise behavioral coding, and the inclusion of multiple retrieval measures allowed for a comprehensive evaluation of how instructional language influences motor memory through both action reproduction and visual identification.

All experimenters involved in data collection completed a standardized training program prior to the study. This training included (1) familiarization with the experimental protocol, including the precise delivery of metaphorical and explicit instructions to ensure consistency across sessions; (2) practice in conducting online sessions, including troubleshooting technical issues (e.g., internet delays and camera adjustments); and (3) review of ethical guidelines for working with child participants, particularly regarding obtaining informed consent and maintaining a child-friendly interaction style.

### 2.4. Data Analyses

In the free recall task, we focused on two dependent variable indicators: the number of actions completed by the children and the mean action scores. The screen-recorded videos of children’s performances were anonymized by removing all identifying information (e.g., names) and labeled with random codes. Two trained raters, who were unaware of which instructional group each participant belonged to, independently scored the movement quality based on the degree of completion using a 5-point scale, targeting three main body parts: the head, upper limbs, and lower limbs. The scoring criteria were adapted from the study by [18] ([18]), as follows: (1) whether the position and orientation of the head matched the original action, assigned 1 point; (2) whether the position of the upper limbs, the direction of both hands, and the orientation of the palms corresponded to the original action, assigned 2 points; and (3) whether the position of the lower limbs and the direction of both feet matched the original action, assigned 2 points. To account for individual differences in children’s flexibility and other factors, strict accuracy regarding the completion angle was not imposed during scoring. The mean action score was calculated by dividing the sum of scores for all completed actions by the number of completed actions. And the final action score for each participant was calculated as the average of the two raters’ scores. The intraclass correlation coefficient (ICC) was used to assess inter-rater reliability. Using a two-way mixed model with consistency type, the results showed good reliability (ICC = 0.81, 95% CI [0.68, 0.89]), indicating good consistency among raters.

For the recognition task, we evaluated children’s recognition accuracy, sensitivity, and confidence levels. Recognition accuracy was calculated by dividing the number of correct identifications by the total number of test items. Sensitivity was assessed using d-prime (d’) values ([13]). Judgment trials were categorized as hits (correctly identifying the target posture when present) or false alarms (incorrectly identifying a non-target interference posture as the target), and hit and false alarm rates were computed accordingly. To quantify confidence, we calculated the low-confidence rate as the proportion of trials with “unsure” responses relative to the total number of trials, reflecting participants’ uncertainty in recognition judgments.

## 3. Results

### 3.1. Free Recall Task

We conducted an analysis of covariance (ANCOVA) with instruction type (metaphorical vs. explicit) as the independent variable and children’s age as the covariate. We verified that the key statistical assumptions underlying this test were met: The Shapiro–Wilk test indicated that residuals were normally distributed for all variables (all *p* > 0.05), confirming the normality assumption; Levene’s test showed no significant differences in variance were found (all *p* > 0.05), satisfying the homogeneity of variance assumption. The results showed that for the number of actions recalled, the main effect of age was significant, *F* (1, 45) = 9.73, *p* = 0.003, η^2^_p_ = 0.18. After controlling for age, the main effect of instruction type remained significant, *F* (1, 45) = 7.50, *p* = 0.009, η^2^_p_ = 0.14. Post hoc comparisons showed that children in the metaphorical instruction group recalled significantly more actions (M = 8.79) than those in the explicit instruction group (M = 6.92), *t* = 2.74, *p* = 0.009, Cohen’s d = 0.791.

For the average quality score of completed actions, the main effect of age was not significant, *F* (1, 45) = 1.57, *p* = 0.217, η^2^_p_ = 0.035, nor was the main effect of instruction type, *F* (1, 45) = 0.35, *p* = 0.56, η^2^_p_ = 0.008. This suggests that while metaphorical instructions increased the number of actions recalled, they did not significantly impact the overall quality of action execution compared to explicit instructions. These differences in recall performance between the two instruction groups are visually presented in Figure 2.

### 3.2. Recognition Task

Prior to conducting ANCOVA analyses, we verified that the key statistical assumptions (normality, homogeneity of variance) underlying these tests were met. For the recognition accuracy, an analysis of covariance (ANCOVA) was conducted with instruction type as the independent variable and child age as the covariate. The main effect of age was not significant, *F* (1, 45) = 1.36, *p* = 0.249, η^2^_p_ = 0.03; the main effect of instruction type was also not significant, *F* (1, 45) = 0.43, *p* = 0.517, η^2^_p_ = 0.01. These results suggest no significant difference in recognition accuracy between the metaphorical and explicit instruction groups.

For discrimination sensitivity, ANCOVA was performed on d-prime (*d′*) values, with instruction type as the independent variable and age as the covariate. The analysis revealed no significant main effect of age, *F* (1, 45) = 1.91, *p* = 0.174, η^2^_p_ = 0.04, and no significant main effect of instruction type, *F* (1, 45) = 0.82, *p* = 0.369, η^2^_p_ = 0.02. Thus, no group differences were found in children’s ability to distinguish previously seen from unseen actions.

For recognition confidence, ANCOVA was conducted on the proportion of low-confidence responses. The results showed no significant main effect of age, *F* (1, 45) = 0.55, *p* = 0.463, η^2^_p_ = 0.012, and no significant main effect of instruction type, *F* (1, 45) = 0.02, *p* = 0.886, η^2^_p_ < 0.001. These findings indicate that children’s subjective confidence levels during recognition did not significantly differ between the metaphorical and explicit instruction conditions. These differences in the recognition task between the two instruction groups are visually presented in Figure 3.

To further assess the null findings from the traditional analysis, a Bayesian ANCOVA was conducted with recognition accuracy as the dependent variable, instruction type as the fixed factor, and age as a covariate. The results showed that the null model best explained recognition accuracy (BF_10_ = 1.000). Adding age, instruction type, or both did not improve model fit (all BF_10_ < 1), and neither predictor showed meaningful evidence of an effect (BF₍incl₎ = 0.500 for age; 0.339 for instruction type). The 95% credible intervals for both factors included zero. Bayesian ANCOVA results for discrimination sensitivity (BF₍incl₎ of instruction group = 0.397) and recognition confidence (BF₍incl₎ of instruction group = 0.288) also showed that the null model was best supported. Neither age nor instruction group has meaningful effects—all 95% credible intervals include 0, and BF_10_ < 1 for models with these factors. These results suggest that neither age nor instructional type influenced these recognition indicators, providing moderate evidence for the absence of group differences.

## 4. Discussion

The present study found that children who learned with metaphorical instructions completed a greater number of actions overall during the free recall task compared to those who received explicit instructions. Importantly, there was no significant difference between the two groups in the mean scores of action completion, indicating that the metaphorical group not only performed more actions but did so without compromising quality. However, in the recognition task, no significant differences were found between the groups in terms of correctness, discrimination sensitivity, or confidence level.

### 4.1. The Advantage of Metaphorical Instructions in Free Recall Tasks

Metaphorical instruction appears particularly effective in enhancing children’s performance in free recall tasks by reducing cognitive load. The effect size observed in the free recall task indicates a moderate practical impact, suggesting that metaphorical instruction may serve as a meaningful tool for enhancing short-term motor memory in young learners, particularly when recall rather than recognition is prioritized.

Unlike explicit instruction, which typically requires learners to retain and process multiple discrete rules or sequential steps, metaphorical instruction condenses complex motor or cognitive procedures into a single, image-based representation. This simplification eases the burden on working memory—a cognitive resource still maturing in children—allowing them to focus more effectively on executing and retrieving learned actions.

Research by [24] ([24]) supports this view, showing that analogy-based instructions improved both motor performance and memory recall, particularly in tasks requiring procedural integration. Metaphorical cues served as cognitive scaffolds that reduced the need for conscious monitoring and rule-based reasoning, aligning well with children’s limited executive control. Thus, metaphors not only render abstract or unfamiliar content more concrete but also leverage embodied cognition to facilitate memory encoding and retrieval—especially critical in cognitively demanding tasks like free recall. Additionally, [28] ([28]) demonstrated that analogy-based instructions can accelerate motor learning by reducing cognitive load and improving movement efficiency. Their findings further support the notion that metaphorical language can serve as an instructional scaffold, facilitating both the accessibility and retention of internal motor representations.

Metaphorical instruction may also enhance learning by promoting deep semantic processing. By encouraging learners to interpret new content through familiar conceptual frameworks, metaphors facilitate the transformation of abstract or arbitrary material into structured, meaningful representations. According to [8] ([8]), repeated exposure to coherent metaphorical structures activates foundational conceptual metaphors, which help organize information into memory schemas grounded in conceptual consistency rather than surface-level linguistic cues. These structured schemas are more easily retained and retrieved, as they provide learners with analogical access pathways rather than requiring rote verbal rehearsal. Such mechanisms are particularly advantageous for young learners or those from linguistically diverse backgrounds, for whom grammatically explicit instructions may pose greater cognitive challenges. Further supporting the connection between cognitive engagement and physical skill development, [5] ([5]) found that resistance training not only improved adolescents’ motor coordination but also enhanced their self-awareness. This evidence complements our argument that metaphorical instructions—by activating both cognitive and embodied pathways—can enrich motor memory by fostering more meaningful and integrated learning experiences.

The home learning environment also plays a crucial moderating role in metaphorical learning. Compared to formal classroom settings, the familiar and low-pressure home environment enhances psychological safety, encouraging children to engage more openly in movement imitation and creative expression. Metaphorical instructions, with their inherently playful and imaginative qualities, resonate with this relaxed atmosphere and reduce anxiety related to performance. This lower affective barrier can improve willingness to experiment with motor actions and, in turn, increase the fluency and accuracy of movement retrieval during free recall tasks.

While metaphorical instruction shows promise, its benefits are likely context-dependent. It may be less effective in learners with limited metaphor comprehension or in tasks where explicit, rule-based precision is required. Over-reliance on metaphor without supporting cues may risk ambiguity or misunderstanding, particularly in young children.

### 4.2. The Performance Difference Between Free Recall and Recognition Tasks

While metaphorical instruction shows distinct benefits in tasks requiring active retrieval, such as free recall, its advantage appears reduced in recognition tasks, where memory retrieval is externally supported by perceptual cues.

In recognition-based tasks—especially those incorporating pictorial stimuli—the consistency and salience of visual cues significantly influence performance outcomes, often diminishing the role of instructional encoding. [11] ([11]) demonstrated that when recognition tasks present highly distinctive or cue-rich stimuli, such as vivid images of actions or postures, the cognitive burden associated with internal retrieval is minimized. In such cases, learners can rely on surface-level perceptual matching rather than internally generated memory traces. As a result, both metaphorically and explicitly instructed children may perform similarly—not because the instructional strategies are equally effective in principle, but because the task structure itself reduces reliance on prior encoding depth. Therefore, although grounded in embodied cognition, metaphorical instructions may offer limited benefits in recognition tasks, where bodily enactment is not required. Their advantage is likely greater in active recall tasks that more directly engage embodied representations.

Moreover, metaphorical instruction is especially beneficial when learners must reconstruct information without external support. It fosters the creation of internal schematic representations that map new information onto familiar domains, enhancing semantic encoding and strengthening memory traces. [4] ([4]) found that such self-constructed mappings significantly outperformed explicit rule-based instruction in improving motor learning and memory, particularly among children and novices. However, in recognition contexts where perceptual cues prompt retrieval, the need for such internal scaffolding is diminished. The task’s external structure compensates for differences in encoding strategy, thereby narrowing the performance gap between instructional groups. This observation underscores the context-dependent nature of metaphorical instruction—its effectiveness is moderated not only by individual learner characteristics but also by task demands and environmental cues.

While recognition accuracy and sensitivity showed no significant group differences, the inclusion of confidence ratings offers a metacognitive perspective on how children experience uncertainty in memory retrieval. The non-difference in low-confidence rates suggests that instructional type did not affect subjective certainty in the recognition task, which may reflect the dominant role of perceptual matching over encoding depth in this task type.

Another plausible interpretation of the null result is that both instructional types may have promoted relatively shallow encoding for visual recognition. Since the task involved static postures and was limited in duration, neither metaphorical imagery nor explicit detail may have led to deeper consolidation, resulting in comparable performance across groups. This interpretation is supported by the Bayesian ANCOVA, which favored the null model and provided no evidence for instructional effects—suggesting that instructional differences did not meaningfully alter encoding depth in the recognition task. Future studies could test this hypothesis by incorporating delayed recognition tests to assess the durability of encoding over time. Additionally, using dynamic video demonstrations instead of static images may increase perceptual richness and facilitate deeper encoding, which could allow instructional effects to manifest more clearly. Including self-reported imagery use or encoding strategy questionnaires may also help clarify how learners process instructional input during recognition tasks.

### 4.3. Limitations and Future Directions

While this study provides initial empirical support for the benefits of metaphorical instructions in enhancing children’s motor learning and memory—especially in free recall tasks—several limitations warrant consideration. First, the absence of a delayed post-test means that the durability of the metaphorical advantage over time remains unknown. Immediate performance gains do not necessarily translate to long-term retention, and future research should incorporate follow-up assessments to evaluate memory stability.

Second, it is worth noting that metaphorical instruction may not be equally effective for all children. Learners with underdeveloped metaphor comprehension may misinterpret figurative cues, which could hinder rather than help performance. The current study did not examine the role of individual differences, such as children’s language ability, metaphor comprehension ability, cognitive flexibility, or prior exposure to metaphorical language. These factors may significantly moderate the efficacy of metaphor-based instruction.

And there is a lack of formal pilot testing to validate the comprehensibility of the metaphorical instructions among targeted children. Not all children may interpret metaphorical instructions as intended. Inappropriate or overly abstract metaphors may increase cognitive load or lead to misinterpretation. And not all metaphorical instructions may have been equally effective. Variations in metaphor quality—including cultural familiarity, concreteness, and clarity—could influence how well children understood and applied the imagery. Future studies should include pre-testing of instructional metaphors for comprehensibility and cultural fit to improve instructional consistency.

Additionally, the use of online recruitment may have introduced sampling bias, favoring participants from digitally literate households. This may limit the ecological validity of the findings, particularly when considering the diverse educational contexts in which metaphorical instruction might be applied. And although the home environment may provide greater comfort and familiarity, the task design—centered on learning movements from static images via video call—remains a simplified and controlled version of real-world motor learning. Thus, the ecological validity of this study is partial and should be interpreted with caution.

Future research should address current limitations by incorporating delayed post-tests to assess long-term retention and by exploring individual learner factors—such as metaphor comprehension and cognitive style—that may moderate instructional effects. A structured pilot test should be carried out to refine instructions involving larger samples and validated comprehension metrics (e.g., coding verbal explanations or movement accuracy). Comparative studies across online and in-person settings can help clarify the context sensitivity of metaphorical instruction. Additionally, examining the role of metaphor characteristics (e.g., familiarity, vividness, and cultural relevance) and employing process-tracing methods like eye-tracking or neurocognitive assessments will provide deeper insight into the real-time cognitive mechanisms that differentiate metaphorical from explicit instruction. Taken together, these future directions will deepen our understanding of how and when metaphorical instruction can be most effectively deployed to enhance children’s motor learning and memory in both educational and applied settings.

The online implementation of metaphorical instruction in this study offers preliminary insights into how digital platforms can support embodied learning in early childhood. Unlike traditional classroom or sports settings, online environments present both challenges (e.g., reduced physical supervision and technical constraints) and opportunities (e.g., individualized pacing and home-based familiarity). Our findings suggest that metaphorical instruction can remain effective even in remote formats. This has practical implications for the design of remote physical education programs and motor skill interventions, particularly for young learners who may benefit from playful, intuitive forms of instruction in digital settings.

## 5. Conclusions

The results of this study indicate that, in an online learning context, children who received metaphorical instruction recalled and performed more body posture actions during the free recall phase than those who received explicit instruction, with no decline in action quality. However, this advantage was not evident in the body posture picture recognition task, suggesting that the benefits of metaphorical instruction are more pronounced in tasks requiring active retrieval and may be shaped by the learning environment. These findings also offer practical implications for virtual physical education. Metaphorical instruction may help enhance memory and engagement in remote motor learning, especially when physical demonstration or correction is limited. Educators could consider using developmentally appropriate metaphors to support children’s understanding and recall of complex movements in online settings.

## Figures and Tables

**Figure 1 behavsci-15-01132-f001:**
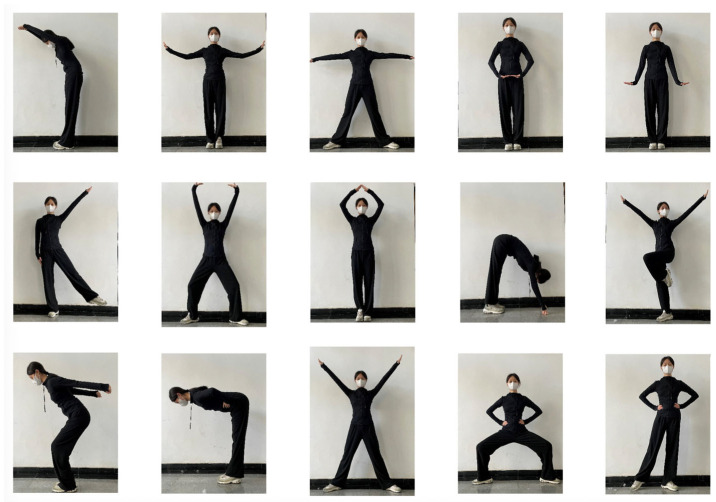
Schematic diagram of 15 movement postures.

**Figure 2 behavsci-15-01132-f002:**
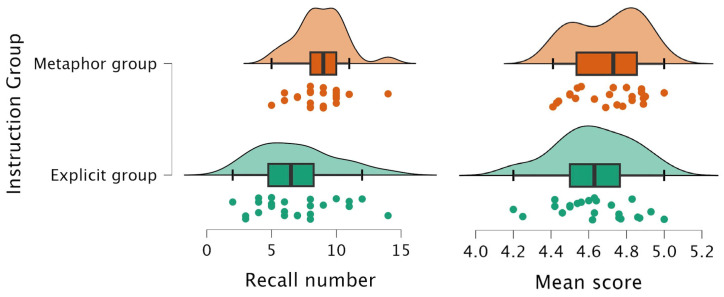
Raincloud plots showing group differences in recalled action quantity (Recall number) and action quality scores (Mean score) across instructional conditions.

**Figure 3 behavsci-15-01132-f003:**
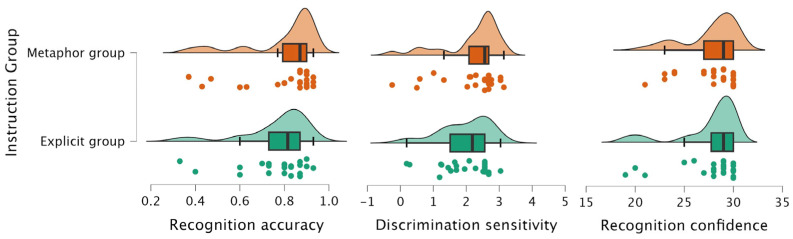
Raincloud plots showing group differences in recognition accuracy, discrimination sensitivity, and recognition confidence across instructional conditions.

**Table 1 behavsci-15-01132-t001:** Examples of metaphorical and explicit instructions for movement postures.

Posture Images	Metaphorical Instructions	Explicit Instructions
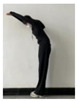	Stand with your feet together, arms straight up, and upper body bent forward like a curved moon.	Stand with your feet together, arms straight up, and upper body leaning forward.
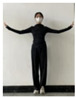	Both arms are raised, and the elbows are lowered as if holding two large balls at the elbows.	Stand straight with your legs together, arms up at shoulder height, elbows down, fingers up.
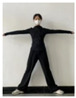	Stand with your feet apart and your arms held flat like the Chinese character “大” (big), with your palms facing down.	Stand with your feet open, arms up at shoulder height, palms facing down.

## Data Availability

The data presented in this study are available on request from the corresponding author. The data are not publicly available due to privacy and ethical restrictions, as the study involved child participants, and we have committed to their guardians that the experimental data would not be made publicly accessible.

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
