# Peer review of "How Metaphorical Instructions Influence Children’s Motor Learning and Memory in Online Settings"

_behavsci, 2025, doi:10.3390/bs15081132_

Round 1
Reviewer 1 Report
Comments and Suggestions for Authors
Thank you for the effort in the development of this manuscript, and congratulations for the work.
Under my point of view, one point for improvement would be related to the organization of the text: The paragraph between lines 38 to 41 could be more appropriately placed immediately after the citation of Shadmehr (2006), currently on page 46, to enhance the logical flow of the argument.
Additionally, within the Methods section, further clarification is needed regarding the decision to include explicit instruction as part of the metaphor-based instructional condition. It would strengthen the rationale if the authors could support this choice with references from previous literature that justify the integration of explicit cues within metaphorical frameworks.
- What is the main question addressed by the research?
The effects of different ways of instruction comparing analogy versus explicit learning.
- Do you consider the topic original or relevant to the field? Does it address a specific gap in the field?
Absolutely, the way to teach to young people if necessary, especially in that age where motor learning it is creating the motor behavior. The evidence about teaching forms it is until the date scarce and to know different ways to teach could be useful to get optimal ways to teach.
- What does it add to the subject area compared with other published material?
Other materials analyse simple tasks as jumping or running. This kind of interventions make another type of analysis during complex motor tasks.
- What specific improvements should the authors consider regarding the methodology?
Humbly, they should specify why the use a mixed method during analogy learning evidence based.
- Are the conclusions consistent with the evidence and arguments presented and do they address the main question posed?
Discussion and conclusion are well argumented presenting a step-by-step model all the results about the intervention.
- Are the references appropriate?
I believe so. In the introduction, they refer to the theoretical foundations of instruction and cueing.
Author Response
Thank you very much for taking the time to review this manuscript. Your detailed comments have provided clear guidance for improving the manuscript, and we will carefully address each point in our revisions. Please find the detailed responses below in the attached file and the corresponding corrections were highlighted in blue in the re-submitted files.

Reviewer 2 Report
Comments and Suggestions for Authors
Thank you for your submission. Your manuscript addresses a timely and important topic. To further strengthen your paper, please consider addressing the following points:
- Explicitly state your hypotheses in the introduction for greater clarity.
- Clearly justify the use of ANCOVA by confirming that key statistical assumptions (normality, homogeneity of variance) were met.
- Provide details about experimenter training, clarify if scoring was blinded, and specify the randomization method used.
- Indicate if metaphorical instructions were pilot-tested with children to ensure comprehension.
- Address how potential variability introduced by the online testing environment (camera angles, visibility, internet delays) was controlled.
- Clarify demographic information, explicitly noting if participants were typically developing.
Addressing these minor points will improve the methodological rigor and clarity of your manuscript. Thank you again for your valuable contribution.
Author Response
Thank you very much for your valuable time and insightful suggestions on our manuscript. We have taken your suggestions seriously and diligently worked on refining the manuscript. Please find the detailed responses below in the attached file and the corresponding revisions were highlighted in blue in the re-submitted files.

Reviewer 3 Report
Comments and Suggestions for Authors
Overall, the research question is worthwhile and of applied significance, but key improvements are required before the work can be considered for publication. Detailed comments in the attachment.

Author Response
Thank you very much for your valuable time and constructive feedback on our manuscript. We are particularly encouraged by your recognition that our work addresses a timely and important topic. Your detailed suggestions have provided clear and insightful guidance for strengthening the paper, and we fully appreciate the importance of addressing each point you raised. Please find the detailed responses below in the attached file and the corresponding revisions were highlighted in blue in the re-submitted files.

Round 2
Reviewer 1 Report
Comments and Suggestions for Authors
Congratulations on this optimizated version of the manuscript, and thank you for carefully considering the suggested revisions. Your study is rigorous, grounded in relevant literature, and clearly presented. The improvements introduced have enhanced the overall clarity and consistency of the work. The methodological approach remains strong, and the theoretical rationale is well supported. I appreciate your attention to detail and responsiveness in addressing the feedback provided. The revised version reflects these efforts and presents a solid contribution to the field.
Reviewer 3 Report
Comments and Suggestions for Authors
Dear Authors,
Thank you for your revised manuscript. You have done a very good job addressing the comments, and I appreciate the way in which you incorporated all of my suggestions. The manuscript has improved significantly and is now much clearer and more convincing.